# Immunohistochemical Screening of HER2 in Canine Carcinomas: A Preliminary Study

**DOI:** 10.3390/ani11041006

**Published:** 2021-04-03

**Authors:** Barbara Brunetti, Barbara Bacci, Giuseppe Sarli, Elisa Pancioni, Luisa Vera Muscatello

**Affiliations:** Department of Veterinary Medical Sciences, University of Bologna, Ozzano dell’Emilia, 40064 Bologna, Italy; b.brunetti@unibo.it (B.B.); giuseppe.sarli@unibo.it (G.S.); elisa.pancioni@libero.it (E.P.); luisaver.muscatello2@unibo.it (L.V.M.)

**Keywords:** HER2, ErbB2, canine, carcinoma, intestine, rectum, squamous cell carcinoma

## Abstract

**Simple Summary:**

The human epidermal growth factor receptor-2 (HER2) has been studied in many human carcinomas. This receptor can be amplified and overexpressed in tumoral cells and serve as a prognostic, therapeutic, and predictive biomarkers. In human medicine there are many target drugs direct against this receptor, and these drugs are able to improve the clinical outcome of the patients. The overexpression of this receptor is mainly evaluated by immunohistochemistry, a technique that allows to evaluate the expression of the receptor on the cell surface. Recently, HER2 expression has been investigated in several canine tumors, but its role in the development of canine carcinoma is an object of debate. In this study we wanted to investigate the expression of HER2 protein in different histotypes of canine carcinomas in order to identify potential tumors that could benefit from the HER2-targeted therapy. We confirmed the hypothesis that HER2 is involved in tumor development of several canine carcinomas, amongst which carcinomas of the intestinal tract predominate. In dogs included in this study, 80% of intestinal carcinomas and a high proportion of rectal carcinomas (42%) were HER2 positive, suggesting that tumors of intestinal origin may potentially benefit from HER2-directed therapy.

**Abstract:**

HER2 overexpression has been reported in various human and canine tumours. The aim of this study is to investigate the expression of HER2 protein in different histotypes of canine carcinomas in order to identify potential tumours that could benefit from the HER2-targeted therapy. Eighty-two (82) canine carcinomas (squamous cell, gastro-intestinal, rectal, pulmonary, prostatic, urothelial, and ovarian) from paraffin-embedded samp les were immunohistochemically evaluated. The degree of HER2 expression was scored based on the ASCO/CAP 2018 guidelines. Intestinal carcinomas were those with greater HER2 overexpression (3+) with 81% of positive cases, followed by 42% of rectal carcinomas and 28% of squamous cell carcinomas. These observations suggest that HER2 overexpression could be a driver in the oncogenesis of several types of canine carcinomas and lay the foundations for the identification of different types of canine carcinomas that could benefit from HER2-targeted therapy.

## 1. Introduction

The canine homolog of human epidermal growth factor receptor (HER2), erb-b2 receptor tyrosin kinase 2 (ErbB2), is a tyrosine-kinase receptor that is essential for cellular proliferation and differentiation, and whose role is widely recognized in the development of several human and canine carcinomas [1]. Oncogenic activation of HER2 commonly occurs through gene amplification, which results in protein overexpression on the cell membrane, leading to transduction of signals that regulate cell growth [1]. In humans, overexpression or amplification of HER2 has been reported in 13–20% of breast cancers [2,3], 7–34% of gastric cancers [4], 3–47% of colorectal cancer [5], and 1.9–14.3% of lung carcinomas [6]. Similarly, its role in carcinogenesis has been investigated in several canine tumours, including mammary [7], urothelial [8], apocrine sac gland [9], lung [10], and thyroid carcinomas [11].

Most of HER2 research has been undertaken in human breast cancer and consequently, in canine mammary tumours. In women, monitoring of tumour HER2 status, including protein overexpression and gene amplification, has become a routine test that is necessary for clinical decision-making process [2]. In fact, positive HER2 status is detected in up to 20% of breast tumours and is associated with poorer prognosis, more aggressive behavior, and increased risk of disease recurrence [2]. Furthermore, knowledge of HER2 status is essential to assess eligibility to HER2-targeted therapy such as with trastuzumab (a humanized monoclonal antibody against the extracellular portion of the HER2 protein, used to treat breast cancer and stomach cancer), which is known to increase overall survival [2]. In dogs, overexpression rates in malignant mammary tumours range from absent [12,13] to as high as 10% [7] and if identified, its prognostic role remains controversial [14,15,16,17]. Clinical trial results are not yet available in dogs with mammary carcinomas, however, growth inhibition of canine carcinoma cell lines has been achieved with trastuzumab in a recent study [18].

The success of trastuzumab therapy in different types of human cancer prompted us to investigate HER2 status in other canine malignancies. Therefore, the aim of the study was to investigate the expression of HER2 by immunohistochemistry, in a spectrum of canine carcinomas of different origin.

## 2. Materials and Methods

Eighty-two (82) cases with a diagnosis of carcinoma were retrieved from the archives of the Pathology section, Department of Veterinary Medical Sciences, University of Bologna and of the Veterinary Hospital “I Portoni Rossi”, Bologna. In detail, carcinomas were divided as follows: intestinal (*n* = 15), rectal (*n* = 14), cutaneous squamocellular (cSCC) (*n* = 14), oral squamocellular (oSCC) (*n* = 8), ovarian from surface epithelial structures (*n* = 7), prostatic (*n* = 5), pulmonary (*n* = 4), renal (*n* = 4), and gastric (*n* = 3). Samples were available as formalin-fixed paraffin-embedded (FFPE) material and routinely stained haematoxylin and eosin (HE) histological sections. One to five slides were available for each neoplasm based on the tumor size. The diagnoses were revised by two board-certified pathologists (B.Br. and L.V.M.), and the appropriate and most recent WHO classification system was applied. Three samples of normal tissues of stomach, small intestine, rectum, skin, oral mucosae, urothelium, ovary, prostate, lung, and kidney were used to investigate the expression of HER2 in normal tissues.

Immunohistochemistry was performed with HER2 antibody known to cross-react with the canine species (polyclonal, Dako, Glostrup, Denmark, dilution 1:200) [19]. Serial sections 3-micron-thick were dewaxed and rehydrated. Endogenous peroxidase was blocked by immersion in 3% H_2_O_2_ in methanol for 30 min at room temperature. Antigen retrieval was performed in a microwave oven at 750 W by incubation in citrate buffer at pH 6.0 for 10 min. The antibody was incubated with the tissue sections overnight at 4 °C. Binding sites were revealed by secondary biotinylated antibody (dilution 1:200) and amplified using a commercial avidin-biotin-peroxidase kit (VECTASTAIN^®^ ABC Kits, Peterborough, UK). The chromogen 3,3′-diaminobenzidine (0.05% for 3 min at room temperature) was used. Slides were counterstained with Papanicolaou’s hematoxylin. A canine mammary carcinoma with a known 3+ positivity was used as positive control. The primary antibody was omitted in the negative control.

HER2 positivity was evaluated following the 2018 ASCO/CAP guidelines [2] where 3+ positive score is assigned with a complete and intense circumferential membrane staining in >10% of tumour cells, 2+ equivocal score is assigned with a weak to moderate, complete membrane staining in >10% of tumour cells; and 1+ negative is assigned with incomplete membrane staining that is barely perceptible and in >10% of tumour cells. When no staining is observed or when there is a membrane staining that is incomplete and faint in <10% of tumour cells, the tumour is considered 0 negative. For intestinal and gastric tumours, immunohistochemical evaluation was performed following guidelines as indicated by 2016 ASCO/CAP guidelines, in which basolateral positivity is accepted instead of complete ring positivity [20]. The evaluations were performed using a Nikon Eclipse E600 with 10× objective.

## 3. Results

Results are summarised in Table 1. Of 15 intestinal carcinomas, 12 (80%) were positive, 2 (13.3%) were 2+ equivocal, and 1 (6.6%) was negative. Of these, all tumours with small intestinal origin (*n* = 7) were positive, while those originating from the colon (*n* = 8) showed variable HER2 scores. Of 14 rectal carcinomas, 6 (42%) were 3+ positive (Figure 1A), 5 (35.7%) were 2+ equivocal, and 3 (21.4%) were 1+ or 0 negative. Normal intestinal mucosa had an intense basolateral HER2 expression. Most (60%) intestinal carcinomas had a markedly heterogeneous intratumoural positivity. There were three gastric carcinomas of which one (33.3%) had a 3+ positivity (Figure 1C). Of 14 cutaneous SCC, 4 (28.5%) were 3+ positive, 2 (14.29%) were 2+ equivocal, and 8 (57.1%) were 1+ or 0 negative. Of eight oral squamous cell carcinomas, three (37.1%) were 3+ positive (Figure 1B), one (12.8%) was 2+ equivocal, and four (40%) were 1+ or 0 negative. Within urothelial carcinomas (UC), six (75%) were 3+ positive (Figure 1D), one (12.5%) was 2+ equivocal, and one (12.5%) was 1+ negative. Three out of five prostatic carcinomas (60%) were 3+ positive (Figure 1E). Two of the UC positive cases displayed heterogeneous intratumoural positivity. Of seven ovarian carcinomas, two (28.5%) were 3+ positive (Figure 1F), two (28.5%) were 2+ equivocal, and three (33%) were 1+ or 0 negative. Within pulmonary and renal carcinomas, two (50%) were 3+ positive while the remaining two (50%) were 1+ or 0 negative. In the normal tissues (stomach, small intestine, rectum, skin, oral mucosae, urothelium, ovary, prostate, lung and kidney), the HER2 expression was cytoplasmic to mild complete membranous or moderate incomplete membranous.

## 4. Discussion

Amplification and/or overexpression of HER2 has been demonstrated in several human and canine tumours. In this study we confirmed the hypothesis that HER2 is involved in tumour development of several canine carcinomas (intestinal, rectal and urothelial), amongst which carcinomas of the intestinal tract predominate. In dogs included in this study, 80% of intestinal carcinomas and a high proportion of rectal carcinomas (42%) were HER2 positive, suggesting that tumours of intestinal origin may potentially benefit from HER2-directed therapy. Similarly, in human colorectal carcinoma, HER2 has been found to be overexpressed. In fact, *HER2* gene amplification is associated with a higher invasiveness and metastatic potential, but despite HER2’s prognostic role remaining controversial, therapy with trastuzumab and lapatinib (an orally active drug for breast cancer and other solid tumours) was found to improve outcome in a subset of cases [5]. Focused studies on canine intestinal carcinomas are needed to correlate HER2 overexpression with tumour behavior, prognosis, and response to specific targeted therapy. A consensus for evaluation system of HER2 is reached only for canine mammary carcinoma [17] but common guidelines on the evaluation of others carcinoma are lacking. Another critical point is the preanalytical variables that may affect the results. Only three cases of gastric carcinoma were included in this study, of which only one was positive, thus a larger caseload needs to be investigated to provide significant data for carcinomas at this location. While using a different scoring system, Terragni et al., have previously investigated HER2 in canine carcinomas and found that 58% of canine gastric carcinomas overexpressed the receptor [21].

Here, HER2 expression was also assessed in canine squamous cell carcinomas, of either oral (oSCC) or cutaneous (cSCC) origin. Compared with carcinomas of intestinal origin, only a minority of SCC was found to overexpress HER2. In particular, only 37% of oSCC and 28% of cSCC of had 3+ positive score. Similar proportions were found in human head and neck squamous cell carcinomas [22,23], suggesting only a limited role of HER2 in this type of malignancy.

HER2 expression in canine urothelial carcinomas (UC) has been previously investigated by immunohistochemistry [8,19] and polymerase chain reaction (PCR) [24]. Compared to normal urothelium, which was negative in all studies, UC was positive in a high percentage of cases, which ranged from 56% of UC in one study [8] to 60.9% in the other study [19]. In the present case series, an even higher proportion of cases was positive (75%), confirming the involvement of HER2 in UC carcinogenesis. Limited information is available on the prognostic role of HER2 in canine UC. In one study, although there was no significant association of HER2 positivity and outcome, HER2 positive cases tended to have a shorter disease-free interval [19].

Variable proportions of ovarian, prostatic, pulmonary, and renal carcinomas also overexpressed HER2, but the small number of cases investigated limits the interpretation of the results.

Intratumoural heterogeneity of positivity has been previously observed in human breast cancer and may indicate the presence of subclones of cells arising from random genetic alterations [25]. Tumour heterogeneity represents a potential confounder for HER2 scoring and highlights those cases that warrant further investigation by Fluorescence in situ hybridization (FISH) analysis. In the present case, series heterogeneity has been observed in a proportion of intestinal tumours and in a minority of UC, suggesting that confirmation of genetic amplification is needed in future studies.

## 5. Conclusions

In summary, in addition to mammary, urothelial, thyroid, and apocrine gland sac carcinomas, we demonstrated that HER2 is overexpressed in a large proportion of intestinal, rectal and urothelial carcinomas, as well as in a subset of oral and cutaneous squamous cell carcinomas. Additional FISH analysis is needed to confirm the presence of HER2 gene amplification behind protein overexpression in these tumours. In conclusion, this preliminary study opens the way for further investigating the prognostic role of HER2 in different types of canine carcinomas and to potentially expand the range of tumours that may benefit from a HER2-targeted therapy.

## Figures and Tables

**Figure 1 animals-11-01006-f001:**
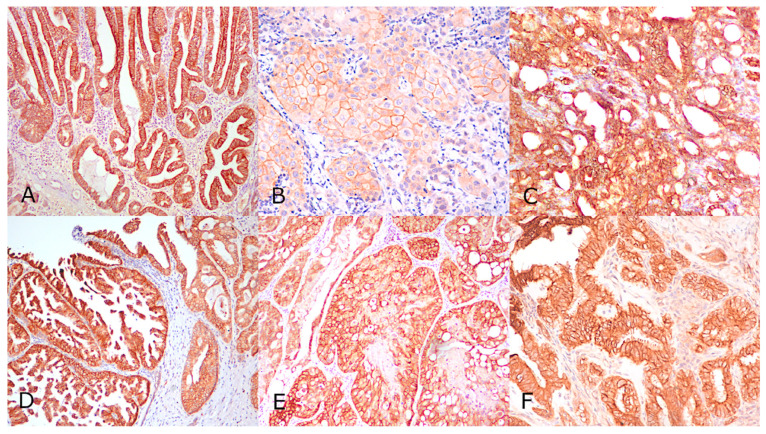
HER2 positivity in canine carcinomas, all cases displaying 3+ positivity characterized by complete and intense circumferential staining of the cell membrane. (**A**) Rectal carcinoma, (**B**) Oral Squamous cell carcinoma, (**C**) Gastric carcinoma; (**D**) Urothelial transitional cell carcinoma, (**E**) Prostatic carcinoma; (**F**) ovarian carcinoma. Avidin-biotin-peroxidase method, counterstained with Papanicolaou’s hematoxylin, 10×.

**Table 1 animals-11-01006-t001:** HER2 expression in canine carcinomas based on location and positivity score.

Tumor	n. Cases	3+ (Positive)	2+ (Equivocal)	1+ (Negative)	0 (Negative)
Intestinal	15	12 (80%)	2 (13.3%)	1 (6.6%)	0
Rectal	14	6 (42.8%)	5 (35.7%)	1 (7.14%)	2 (14.2%)
cSCC	14	4 (28.5%)	2 (14.3%)	2 (14.3%)	6 (42.8%)
oSCC	8	3 (37.2%)	1 (12.8%)	3 (37.5%)	1 (12.5%)
Urothelial	8	6 (75%)	1 (12.5%)	1 (12.5%)	0
Ovarian	7	2 (28.5%)	2 (28.5%)	2 (28.5%)	1 (14.3%)
Prostatic	5	3 (60%)	0	1 (20%)	1 (20%)
Pulmonary	4	2 (50%)	0	1 (25%)	1 (25%)
Renal	4	2 (50%)	0	0	2 (50%)
Gastric	3	1 (33.3%)	1 (33.3%)	0	1 (33.3%)
Total	82	41	14	12	15

cSCC: cutaneous squamous cell carcinoma, oSCC: oral squamous cell carcinoma.

## Data Availability

The data presented in this study are available on request from the corresponding author. The data are not publicly available due to privacy.

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
