# Peer review of "Immunohistochemical Screening of HER2 in Canine Carcinomas: A Preliminary Study"

_animals, 2021, doi:10.3390/ani11041006_

Round 1

Reviewer 1 Report

The manuscript “Immunohistochemical screening of HER2 in canine carcinomas” “was an assessment by immunohistochemistry of HER2 expression in the several canine carcinomas.

Although not innovative it is an interesting work because it intends to provide data for the identification of different types of canine carcinomas that could benefit of HER2-targeted therapy. The work is well explained, with sections presented in a balanced and coherent way the data are properly discussed. I suggest Accept after minor revision.

TITLE

Comment 1: As the authors refer in the Conclusions: “…this preliminary study…” (lines 171-172) I suggest: “Immunohistochemical screening of HER2 in canine carcinomas: a preliminary study”.

Comment 2: At the beginning of the manuscript,  I suggest presenting a list of the abbreviations, I found these:

cSCC: cutaneous squamous cell carcinomas

ErbB2: erb-b2 receptor tyrosin kinase 2

FFPE: Formalin-Fixed Paraffin-Embedded

FISH: Fluorescence in situ hybridization

HE: haematoxylin and eosin

HER2: human epidermal growth factor receptor

oSCC: oral squamous cell carcinomas

PCR: Polymerase chain reaction

UC: urothelial carcinomas

INTRODUCTION

Comment 3: As the work is about the expression of HER2 I think it is better in line 37 to start the Introduction with HER2, for what I suggest, one of two:

“The canine homolog of HER2 (human epidermal growth factor receptor; erb-b2 receptor tyrosin kinase 2) is a tyrosine-kinase receptor…”

“The canine homolog of human epidermal growth factor receptor (HER2), the erb-b2 receptor tyrosin kinase 2 (ErbB2) is a tyrosine-kinase receptor…”

Comment 4 (lines 55-56): I suggest “…trastuzumab (a humanized monoclonal antibody against the extracellular portion of the 55 HER2 protein, used to treat breast cancer and stomach cancer),…”

MATERIALS AND METHODS

Comment 5: Authors should mention the magnification used to make the evaluation by optical microscopy and how the photomicrographs were captured.

Comment 6: The authors should mention how many slides were observed per animal, and how they were observed: only once by an investigator, or by more than one investigator, independently?

RESULTS

Comment 7 (Table 1): In tumors appears OSCC but only in line 142 the authors explain it is an oral squamous cell carcinoma. Or refer in line 70 as they do in line 69 for cutaneous squamocellular (cSCC) or at the bottom of the table refer to the meaning of both acronyms, deleting cSCC in line 69. One the other hand should standardize with lowercase and uppercase letters or only with uppercase letters.

Comment 8 (Figure 1 caption). If the primary antibody used was HER2 then it should be  “Figure 1. HER2 positivity in canine carcinomas…” and not “Figure 1. ErbB2 positivity in canine carcinomas…”

Comment 9 (Figure 1 caption): At the end I suggest add one of these:

Avidin-biotin-peroxidase method, counterstained with Papanicolaou’s hematoxylin. Magification/Scale bar xx micron.

Counterstained with Papanicolaou’s hematoxylin. Magification/Scale bar xx micron.

DISCUSSION

Comment 10 (line 133): As in the Introduction the authors said what trastuzumab is, I suggest do the same for lapatinib, so I suggest:

"…with trastuzumab and lapatinib (an orally active drug for breast cancer and other solid tumours), was found to improve outcome in a subset of cases [5]”.

Author Response

The manuscript “Immunohistochemical screening of HER2 in canine carcinomas” “was an assessment by immunohistochemistry of HER2 expression in the several canine carcinomas.

Although not innovative it is an interesting work because it intends to provide data for the identification of different types of canine carcinomas that could benefit of HER2-targeted therapy. The work is well explained, with sections presented in a balanced and coherent way the data are properly discussed. I suggest Accept after minor revision.

TITLE

Reviewer: Comment 1: As the authors refer in the Conclusions: “…this preliminary study…” (lines 171-172) I suggest: “Immunohistochemical screening of HER2 in canine carcinomas: a preliminary study”.

Comment 2: At the beginning of the manuscript,  I suggest presenting a list of the abbreviations, I found these:

cSCC: cutaneous squamous cell carcinomas

ErbB2: erb-b2 receptor tyrosin kinase 2

FFPE: Formalin-Fixed Paraffin-Embedded

FISH: Fluorescence in situ hybridization

HE: haematoxylin and eosin

HER2: human epidermal growth factor receptor

oSCC: oral squamous cell carcinomas

PCR: Polymerase chain reaction

UC: urothelial carcinomas

Authors: considering that is not in the guidelines of the journal an abbreviations section, we prefer to add the single abbreviation in the manuscript.

INTRODUCTION

Reviewer: Comment 3: As the work is about the expression of HER2 I think it is better in line 37 to start the Introduction with HER2, for what I suggest, one of two:

“The canine homolog of HER2 (human epidermal growth factor receptor; erb-b2 receptor tyrosin kinase 2) is a tyrosine-kinase receptor…”

“The canine homolog of human epidermal growth factor receptor (HER2), the erb-b2 receptor tyrosin kinase 2 (ErbB2) is a tyrosine-kinase receptor…”

Authors: Done.

Reviewer: Comment 4 (lines 55-56): I suggest “…trastuzumab (a humanized monoclonal antibody against the extracellular portion of the 55 HER2 protein, used to treat breast cancer and stomach cancer),…”

Authors: Done

MATERIALS AND METHODS

Reviewer: Comment 5: Authors should mention the magnification used to make the evaluation by optical microscopy and how the photomicrographs were captured.

Authors: The following sentence was added in the material and methods:

“The evaluation were performed using a Nikon Eclipse E600 with 10x objective”.

Reviewer: Comment 6: The authors should mention how many slides were observed per animal, and how they were observed: only once by an investigator, or by more than one investigator, independently?

Authors: The following sentence has been added in the text:

“One to five slides were available for each neoplasm based on the tumor size. The diagnosis were revised by two board-certified pathologists (B.Br and L.V.M.).

RESULTS

Reviewer: Comment 7 (Table 1): In tumors appears OSCC but only in line 142 the authors explain it is an oral squamous cell carcinoma. Or refer in line 70 as they do in line 69 for cutaneous squamocellular (cSCC) or at the bottom of the table refer to the meaning of both acronyms, deleting cSCC in line 69. One the other hand should standardize with lowercase and uppercase letters or only with uppercase letters.

Authors: Done.

Reviewer: Comment 8 (Figure 1 caption). If the primary antibody used was HER2 then it should be  “Figure 1. HER2 positivity in canine carcinomas…” and not “Figure 1. ErbB2 positivity in canine carcinomas…”

Authors: Done

Reviewer: Comment 9 (Figure 1 caption): At the end I suggest add one of these:

Avidin-biotin-peroxidase method, counterstained with Papanicolaou’s hematoxylin. Magification/Scale bar xx micron.

Counterstained with Papanicolaou’s hematoxylin. Magification/Scale bar xx micron.

Authors: Done

DISCUSSION

Reviewer: Comment 10 (line 133): As in the Introduction the authors said what trastuzumab is, I suggest do the same for lapatinib, so I suggest:

"…with trastuzumab and lapatinib (an orally active drug for breast cancer and other solid tumours), was found to improve outcome in a subset of cases [5]”.

Authors: Done.

Reviewer 2 Report

The work entitled “Immunohistochemical screening of HER2 in canine carcinomas” gives a new insight on Her2 expression in canine carcinomas of different origin. This manuscript is well written and tackles an important topic. My biggest concern associated to this work is that the HER evaluation was only performed by immunohistochemistry and was performed in a low number samples of some types of tumors that limits the analysis and conclusions.

Could the authors clarify why FISH analysis was not performed?

If the authors could increase the number of tumors and/or perform FISH analysis the reviewed version of this manuscript could be considered for publication as full article. If not, I think with the current results this work can be at most considered for Short Communication.

Author Response

Reviewer: The work entitled “Immunohistochemical screening of HER2 in canine carcinomas” gives a new insight on Her2 expression in canine carcinomas of different origin. This manuscript is well written and tackles an important topic. My biggest concern associated to this work is that the HER evaluation was only performed by immunohistochemistry and was performed in a low number samples of some types of tumors that limits the analysis and conclusions.

Could the authors clarify why FISH analysis was not performed?

If the authors could increase the number of tumors and/or perform FISH analysis the reviewed version of this manuscript could be considered for publication as full article. If not, I think with the current results this work can be at most considered for Short Communication.

Authors: The aim of the study was to immunohistochemically evaluate the protein expression of HER2 in different types of carcinoma. We would like to investigate the HER2 amplification with fluorescence in situ hybridization in the next study. We collected the maximum number of cases in our database, therefore we could not increase the number.

Reviewer 3 Report

The paper describes potential role and use of prognostic factor of HER2. The paper is well written and easy to read. Authors measured expression of HER2 in number of samples. The main drawback of the paper is information regarding the expression in healthy tissue. Author cited paper where normal urothelium  cells were negative for HER2. This information (taken form the literature) is required for other tissues investigated what would serve as a negative control. Authors may also perform the experiments collecting tissue post mortem. This is the main reason why I am suggesting major review. Other parts of MS are well written and appropriately discussed.  

Authors wrote that “A canine mammary carcinoma with a known 3+ positivity was used as positive control”. The information regarding origin of the control should be provided. The information also should include how the assessment of the expression in the positive control was performed.  Was it assessed also by the procedure described by Authors. This is important since when developing new method some “golden standard” should be used. Summing up, how come Authors are sure that the positive sample was really 3+ for HER2.

Authors wrote: “...we confirmed the hypothesis that HER2 is in-125 volved in tumour development of several canine carcinomas, amongst which carcinomas of the intestinal tract predominate” . This may be true for Intestinal , Rectal or Urotherial  but the conclusions are not so clear regarding other cannot be drown since too small groups. Further in text Authors discuss this issue, but this sentence needs to be rewritten and indicate which carcinomas may be associated with HER2 overexpression.

Line 15. „In this study we want to investigate” switch to „In this study we wantED to investigate”

Line 78 “H2O2” switch to „H2O2

Author Response

Reviewer: The paper describes potential role and use of prognostic factor of HER2. The paper is well written and easy to read. Authors measured expression of HER2 in number of samples. The main drawback of the paper is information regarding the expression in healthy tissue. Author cited paper where normal urothelium  cells were negative for HER2. This information (taken form the literature) is required for other tissues investigated what would serve as a negative control. Authors may also perform the experiments collecting tissue post mortem. This is the main reason why I am suggesting major review. Other parts of MS are well written and appropriately discussed.  

 Authors: We collected three samples for each tissue (stomach, small intestine, rectum, skin, oral mucosa, urothelium, ovary, prostate, lung, kidney) and we performed additional immunohistochemistry to HER2 on these tissues. We obtained a cytoplasmic to mild complete membranous expression or moderate incomplete membranous expression on all the epithelial cells of the examined tissues”.

Reviewer: Authors wrote that “A canine mammary carcinoma with a known 3+ positivity was used as positive control”. The information regarding origin of the control should be provided. The information also should include how the assessment of the expression in the positive control was performed.  Was it assessed also by the procedure described by Authors. This is important since when developing new method some “golden standard” should be used. Summing up, how come Authors are sure that the positive sample was really 3+ for HER2.

 Authors: The HER2 3+ overexpressed mammary was previously corrected in another caseload of mammary carcinoma-only study. The positivity was confirmed with an amplification of HER2 gene tested by fluorescence in situ hybridization (not yet published study).

Reviewer: Authors wrote: “...we confirmed the hypothesis that HER2 is in-125 volved in tumour development of several canine carcinomas, amongst which carcinomas of the intestinal tract predominate” . This may be true for Intestinal , Rectal or Urotherial  but the conclusions are not so clear regarding other cannot be drown since too small groups. Further in text Authors discuss this issue, but this sentence needs to be rewritten and indicate which carcinomas may be associated with HER2 overexpression.

 Authors: We added in that sentence the intestinal rectal and urothelial carcinoma sites.

Reviewer: Line 15. „In this study we want to investigate” switch to „In this study we wantED to investigate”

Authors: Done.

Reviewer: Line 78 “H2O2” switch to „H2O2

Authors: Done.

Reviewer 4 Report

The authors investigated the expression of HER2 by immunohistochemistry in 82 different canine carcinomas. The introduction and discussion are focused on the comparison with human oncology, and the possibility to apply similar target therapy, also in canine cancer. I feel that the main limit in the veterinary practice is the lack of a standardization process for immunohistochemical staining, including both preanalytical and analytical steps, that can severely affect the quality and the intensity of the IHC signal. Thus, the finding of this paper may be considered the first step to explore the significance of HER2 expression in canine cancer, but this is not enough to speculate on target therapy utility. I think that this limit should be better emphasized by the authors in the discussion.  

Author Response

Reviewer: The authors investigated the expression of HER2 by immunohistochemistry in 82 different canine carcinomas. The introduction and discussion are focused on the comparison with human oncology, and the possibility to apply similar target therapy, also in canine cancer. I feel that the main limit in the veterinary practice is the lack of a standardization process for immunohistochemical staining, including both preanalytical and analytical steps, that can severely affect the quality and the intensity of the IHC signal. Thus, the finding of this paper may be considered the first step to explore the significance of HER2 expression in canine cancer, but this is not enough to speculate on target therapy utility. I think that this limit should be better emphasized by the authors in the discussion.  

Authors: We added a brief discussion on this issue in the discussion: "A consensus for evaluation system of HER2 is reached only for canine mammary carcinoma [17] but common guidelines on the evaluation of others carcinoma is lacking. Another critical point is the preanalytical variables that may affect the results"

Round 2

Reviewer 2 Report

From the authors reply I assume they are not willing/able to incorporate new data in the manuscript since they state that HER2 amplification with fluorescence in situ hybridization in the will be perform in next study and they collected the maximum number of cases in their database. Therefore, I do not feel that this manuscript has enough new information to be published in Animals.

Reviewer 3 Report

Authors explained and replied to the remarks. My recommendation is to publish the paper.